# Comparative Transcriptomics Analysis of Roots and Leaves under Cd Stress in *Calotropis gigantea* L.

**DOI:** 10.3390/ijms23063329

**Published:** 2022-03-19

**Authors:** Jingya Yang, Lingxiong Li, Xiong Zhang, Shibo Wu, Xiaohui Han, Xiong Li, Jianchu Xu

**Affiliations:** 1Yunnan Key Laboratory for Wild Plant Resources, Department of Economic Plants and Biotechnology, Kunming Institute of Botany, Chinese Academy of Sciences, Kunming 650201, China; yangjingya@mail.kib.ac.cn (J.Y.); zhangxiong@mail.kib.ac.cn (X.Z.); wushibo@mail.kib.ac.cn (S.W.); hanxiaohui@mail.kib.ac.cn (X.H.); 2University of Chinese Academy of Sciences, Beijing 100049, China; 3Honghe Center for Mountain Futures, Kunming Institute of Botany, Chinese Academy of Sciences, Honghe 654400, China; 4Germplasm Bank of Wild Species, Kunming Institute of Botany, Chinese Academy of Sciences, Kunming 650201, China; lilingxiong@mail.kib.ac.cn; 5College of Life Sciences, Shaanxi Normal University, Xi’an 710119, China

**Keywords:** cadmium stress, *Calotropis gigantea*, phytoremediation, transcriptome, antioxidant system, metallothionein

## Abstract

*Calotropis gigantea* is often found in mining areas with heavy metal pollution. However, little is known about the physiological and molecular response mechanism of *C. gigantea* to Cd stress. In the present study, Cd tolerance characteristic of *C. gigantea* and the potential mechanisms were explored. Seed germination test results showed that *C. gigantea* had a certain Cd tolerance capacity. Biochemical and transcriptomic analysis indicated that the roots and leaves of *C. gigantea* had different responses to early Cd stress. A total of 176 and 1618 DEGs were identified in the roots and leaves of *C. gigantea* treated with Cd compared to the control samples, respectively. Results indicated that oxidative stress was mainly initiated in the roots of *C. gigantea*, whereas the leaves activated several Cd detoxification processes to cope with Cd, including the upregulation of genes involved in Cd transport (i.e., absorption, efflux, or compartmentalization), cell wall remodeling, antioxidant system, and chelation. This study provides preliminary information to understand how *C. gigantea* respond to Cd stress, which is useful for evaluating the potential of *C. gigantea* in the remediation of Cd-contaminated soils.

## 1. Introduction

Because of industrial activities and excessive use of fertilizers and pesticides, serious cadmium (Cd) pollution has occurred in many areas of the world [1]. As a nonessential element for the growth and development of organisms, Cd can be toxic to plants, animals, and humans, even at low concentrations. Cd can cause a myriad of diseases, including cancer, osteoporosis, renal dysfunction and cardiovascular diseases [2,3]. The threats posed by Cd to humans have garnered significant interest since the emergence of itai-itai disease in Japan in 1955 [4]. In soils, plants are in direct contact with Cd pollution, and edible plants are major sources of Cd entering human bodies via food chains. The toxic symptomatology of Cd on plants mainly presents as growth, photosynthesis and respiration inhibition [5], cell structure destruction and disorder of plant metabolism [6]. However, some plants have evolved a variety of strategies to tolerate and accumulate different concentrations of Cd. It has even been reported that low concentrations of Cd can promote the growth of some plants in a process known as hormesis [7].

Plants have four major tolerance mechanisms against Cd stress: (1) negatively charged substances in the components of the cell wall can adsorb and fix Cd, partially preventing Cd from entering protoplasts [8]; (2) when Cd enters the protoplast, some small molecule organics, such as phytochelatin and metallothionein will be induced to chelate with Cd [9]; (3) some metal transporters can transport Cd into vacuoles with low metabolic activity or that can expel Cd out of cells, which fundamentally alleviates the toxic effect of Cd on cells [9]; and (4) Cd entering the cells activates the antioxidant system to remove excess reactive oxygen species (ROS) accumulation in plants [9]. Plants with high tolerance capacities to Cd can be designated as Cd accumulators/hyperaccumulators to remediate Cd-polluted soils [10]. Over the past few decades, a number of potential phytoremediation resources for Cd have been identified [11]. However, Cd phytoremediation efficiency still needs to be greatly improved because of small biomasses, slow growth rates, and poor environmental inadaptability of many reported heavy metal accumulators possess [12]. Therefore, screening/ breeding Cd accumulators/hyperaccumulators with large biomasses and considerable environmental adaptability for Cd phytoremediation are urgently needed.

*Calotropis gigantea*, which is a fiber and medical plant [13], has strong resistance to drought, saline alkali, and barren conditions [14]. Moreover, *C. gigantea* generally shows a rapid growth rate. These advantages enable *C. gigantea* to have great potential in wind prevention, sand fixation, soil erosion control, and ecological restoration projects [15]. In recent years, the phytoremediation potential of *C. gigantea* for heavy metal pollution has also begun to receive attention. For example, *C. gigantea* has been found to grow well in some power stations, mining areas and other areas polluted by heavy metals in India, Pakistan, and Australia [16,17]. D’Souza et al. (2010) found that *C. gigantea* could absorb up to 2.8 mg kg^−1^ Cd in leaves [18]. Meravi (2014) found that this was a positive correlation in heavy metal concentrations between *C. gigantea* and the environment [19]. Gajbhiye et al. (2019) found that the leaves of *C. gigantea* growing on roadsides could adsorb toxic metals (e.g., Pb, Cd, Cu, Ni, and Zn) from the air, in which Cd concentrations were as high as 5 mg kg^−1^ [20]. These studies indicate that *C. gigantea* has certain tolerances and accumulation characteristics of heavy metals (e.g., Cd) in the natural environment. However, we still lack an accurate understanding of heavy metal tolerance and accumulation characteristics in *C. gigantea* because of the complex and unclear field environmental conditions across different study areas. Moreover, little is known about the physiological and molecular mechanisms behind the response of *C. gigantea* to Cd. In the present study, we performed a seed germination test to assess the Cd tolerance capacity of *C. gigantea* and used comparative transcriptomics technology to explore the molecular response mechanisms of *C. gigantea* roots and leaves to Cd. This study will be useful for evaluating the phytoremediation potential of *C. gigantea* for Cd-polluted soils and/or the feasibility of cultivating *C. gigantea* in Cd-contaminated soils at scale.

## 2. Results

### 2.1. Effect of Cd Treatment on Seed Germination

To understand Cd tolerance of *C. gigantea*, we tested seed germination under different concentrations of Cd solution (Figure 1A). Figure 1B indicates that the seed germination rates did not show significant changes under different Cd treatment concentrations (Figure 1B). However, the root lengths of seedlings were markedly influenced by Cd treatment. As shown in Figure 1C, the root length of *C. gigantea* treated with 5 mg L^−1^ Cd was approximately 44.43% longer (0.001 < *p* < 0.01) than that of the control sample, whereas the root lengths of *C. gigantea* were significantly inhibited (0.001 < *p* < 0.01 or *p* < 0.001) by 15, 20, and 30 mg L^−1^ Cd. The Cd tolerance index of *C. gigantea* showed the same change trends as root length (Figure 1D).

### 2.2. Cd Accumulation and Physiological Changes in Roots and Leaves

Although there was no significant change in the morphology of *C. gigantea* seedlings after 24 h of Cd treatment (Figure 2A), a certain amount of Cd was accumulated in the plants. The average Cd concentrations in roots and leaves were 3.86 and 0.75 mg kg^−1^, respectively, under 30 mg L^−1^ Cd treatment (Figure 2B). In the control group, there was no source of Cd exposure, so no Cd in roots and leaves was detected (Figure 2B).

Under the control condition, there was no significant difference in H_2_O_2_ concentration and CAT activity between the roots and leaves of *C. gigantea*. However, H_2_O_2_ concentrations in roots increased from 3.12 to 13.63 μmol g^−1^ after Cd treatment, whereas those in leaves decreased from 6.07 to 3.12 μmol g^−1^ (Figure 2C). Under Cd treatment, CAT activities in roots decreased from 16.13 to 7.05 U g^−1^ min^−1^, whereas those in leaves increased from 17.52 to 33.32 U g^−1^ min^−1^ (Figure 2D). Overall, the H_2_O_2_ content and CAT activity in both roots and leaves were significantly altered by Cd treatment, but the direction of change differed.

### 2.3. Differential Transcriptional Responses to Cd Stress in Roots and Leaves

To uncover the changes in the expression of genes involved in Cd response, high-throughput RNA-sequencing was performed in roots and leaves. Approximately 7.04–12.05 GB clean data were obtained for 12 samples (Appendix A). The Q20 and Q30 values of the clean data exceeded 97% and 94%, respectively (Appendix A), indicating that the sequencing quality in this study was good. More than 93% of the data measured by the transcriptome were mapped to the genome sequence of *C. gigantea* (unpublished), and the uniquely mapped reads exceeded 91% (Appendix A). A total of 16,453–17,568 genes were identified for 12 samples (Appendix A). 

Based on the screening criteria (fold change ≥ 2, *Padj <* 0.05) for DEGs, a total of 176 (31 upregulated and 145 downregulated) and 1618 DEGs (479 upregulated and 1139 downregulated) were identified in roots and leaves of *C. gigantea* treated by Cd compared to the control samples, respectively (Figure 3A,B). The number of DEGs detected in leaves was approximately 11 times that in roots, suggesting that roots are potentially more insensitive than leaves to Cd. The Venn diagram of DEGs showed that only 47 DEGs were concurrently identified in roots and leaves (Figure 3C), indicating a specificity in response Cd stress in different tissues. To validate the reliability of the RNA sequencing, 12 DEGs with different expression patterns were randomly selected for qRT-PCR (Appendix A). The results showed significant correlations between the data of RNA sequencing and qRT-PCR in both roots (R^2^ = 0.8216, *p <* 0.001) and leaves (R^2^ = 0.9657, *p <* 0.001) (Figure 3D), indicating that reliable expression results obtained by transcriptomic in the present study.

### 2.4. GO Enrichment

DEGs were classified by GO to understand their functions. DEGs related to the molecular function of catalytic activity, binding, transporter activity, TF activity (nuclear acid binding), antioxidant activity, molecular function regulator, molecular transducer activity, electron carrier activity, and structural molecule activity were significantly enriched in roots (Figure 4A). All of these nine GO terms were also enriched in leaves (Figure 4B). Three GO terms, including signal transducer activity, TF activity (protein binding), and translation regulator activity, were specifically enriched in leaves (Figure 4A,B). To better understand the response mechanisms of *C. gigantea* to Cd, we analyzed the following GO terms.

#### 2.4.1. DEGs Involved in Metal or Metal Complex Transporter Genes

There were 7 (downregulated) and 23 (11 upregulated and 12 downregulated) DEGs involved in metal or metal complex transport identified in roots and leaves, respectively (Figure 4C–E; Appendix A). Four DEGs were identified simultaneously in roots and leaves, which showed different change patterns (Figure 4E). Among them, *HKT6* (Asia06G000937) and *DTX42* (Asia02G002256) were downregulated in both roots and leaves, whereas *DTX49* (Asia03G000509) and *IRT1* (Asia05G001745) were downregulated in roots but upregulated in leaves (Figure 4C,D). Six DEGs of ABC transporter family were identified in leaves. All genes belonging to the ABCG subfamily were downregulated, whereas two and one genes of the ABCB subfamily showed up- and down-regulated expression, respectively (Figure 4D).

#### 2.4.2. DEGs Involved in the Antioxidant System

After Cd treatment, 6 and 16 DEGs, which were related to oxidoreductase activity, were identified in roots and leaves, respectively (Figure 4F). We focused on the antioxidant process in the *C. gigantea* response to Cd. Six downregulated DEGs that encode peroxidases (POD), *PER5* (Asia03G002536), *PER24* (Asia10G001543), *PER7* (Asia03G002634), *PER27* (Asia06G000863), *PER60* (Asia07G001846 and Asia07G001847) were identified in roots (Appendix A). In the leaves, 9 DEGs encoding peroxidase and oxidase, 4 DEGs encoding thioredoxins (TRXs), 3 DEGs encoding CATs, 2 DEGs encoding glutathione redoxin, and 1 DEG encoding SOD were identified (Appendix A). Most of the antioxidant enzymes or protein encoding genes in leaves were upregulated, indicating that these upregulated genes may play an important role in controlling the accumulation of excessive reactive oxygen species (ROS) under Cd stress.

#### 2.4.3. DEGs Involved in Cd Chelation 

Under Cd treatment, three metallothionein encoding genes and one defensin protein encoding gene were upregulated in the leaves of *C. gigantea* (Appendix A). These upregulated genes are likely involved in Cd detoxification in *C. gigantea* through the synthesis of chelations (i.e., metallothionein and defensin).

#### 2.4.4. DEGs Involved in Cell Wall Hydrolysis and Repair

Although the DEGs were not enriched in cell wall hydrolysis and repair in the present study, we focused on DEGs involved in this process found in most heavy metal stress studies. Eight DEGs (one upregulated and seven downregulated) related to the cell wall were found in the roots treated with Cd compared with the control sample (Appendix A). There were thirty genes (14 upregulated and 16 downregulated) related to cell wall hydrolysis and repair identified in leaves (Appendix A). 

### 2.5. KEGG Pathway Enrichment 

KEGG pathway enrichment was performed by the hypergeometric test. A total of 7 significantly enriched pathways (excluding metabolic pathways and biosynthesis of secondary metabolites) were identified in both roots and leaves (Figure 5A,B), which shared two common pathways, phenylpropane biosynthesis and flavonoid biosynthesis (Figure 5C). The number of upregulated or downregulated genes in each pathway is shown in Figure 5D,E. In these enriched pathways in roots, all the genes involved in the flavonoid biosynthesis pathway were upregulated, while all the genes involved in the nitrogen metabolism pathway were downregulated (Figure 5D). In leaves, the genes involved in cutin, suberine, and wax biosynthesis were all downregulated, whereas the genes involved in other pathways showed different change trends (Figure 5E).

## 3. Discussion

### 3.1. Cd Tolerance Characteristics of C. gigantea

As a prerequisite for the establishment of plant seedlings, seed germination is often used to test the tolerance of plants to heavy metal stress [21,22,23]. In the present study, the seed germination rates of *C. gigantea* were little affected by Cd stress (Figure 1A,B), which agrees with previous studies [21,23]. This was because the seeds were wrapped in a hard shell, and Cd could not easily enter the cytoplasm, so seed germination is not significantly affected by Cd. Once seeds germinate, plant tissues were directly exposed to Cd, which is vulnerable to Cd. Therefore, after seed germination, root elongation was inhibited with the increase of Cd concentration. Cd treatment had an effect of “low-promotion, high-inhibition” on the root length of *C. gigantea* (Figure 1B), similar to previous studies [24]. Low concentrations of Cd (5 mg L^−1^) could activate the seeds, leading to a significant increase in root length (Figure 1C). This phenomenon is called hormesis [25]. This was consistent with previous studies [21,23,26]. The root lengths of *C. gigantea* were markedly influenced (0.001 < *p <* 0.01 or *p* < 0.001) with increasing Cd concentrations (Figure 1C). To evaluate the Cd tolerance capacity of *C. gigantea*, TIs were calculated on the basis of root lengths. According to the formula [27], the TIs had a similar change trend as root length with increasing Cd treatment concentrations (Figure 1D). Notably, under 15 mg L^−1^ Cd treatment, the TI (> 50%) of *C. gigantea* was still relatively high (Figure 1D), which was similar to the Cd hyperaccumulator *Amaranthus caudatus* under 10 and 20 mg L^−1^ Cd treatments [28]. These results suggest that *C. gigantea* has a certain tolerance capacity to Cd, which agrees with that *C. gigantea* can grow in moderate Cd-contaminated soils for ecological restoration.

When *C. gigantea* plants were treated by Cd for 24 h, the Cd concentrations in roots of *C. gigantea* was approximately 5 times higher than in the leaves, indicating that the plant did not transport Cd efficiently to the leaves within 24 h. The results were different from a previous study that more Cd accumulation in leaves than that in roots of *C. gigantea* grown in Cd-polluted environments over a long period of time [29]. Biochemical analysis showed that CAT activity decreased in roots but increased in leaves under Cd treatment. One possible explanation is that the much Cd absorbed by roots made the roots show a toxic state. Because a small amount of Cd was transported in leaves, the CAT activity was activated to control ROS accumulation. Changes in H_2_O_2_ concentrations in roots and leaves conformed to the changes in CAT activities. These results indicate that different tissues of *C. gigantea* have different responses to Cd, which is related to Cd concentrations in the tissues.

### 3.2. Comparative Transcriptomic Analysis of Root and Leaf Responses to Cd

To systematically understand the responses of roots and leaves of *C. gigantea* to Cd stress, comparative transcriptomic analysis was performed. The number of DEGs found in leaves under Cd treatment was approximately 9 times what was found in roots (Figure 3A,B), although Cd concentrations in roots were higher than that in leaves (Figure 2B). Among these DEGs, the proportion of upregulated genes in leaves was larger than that in roots. These results are likely attributed to the different Cd subcellular distributions in the roots and leaves of *C. gigantea*. We speculated that Cd in roots may be mainly distributed in inactive cell components (e.g., cell wall). Interestingly, most of the potential genes involved in heavy metal detoxification were downregulated in roots but upregulated in leaves, which was consistent with the aforementioned biochemical analysis results.

To better understand the Cd response of *C. gigantea*, the main Cd transport and detoxification processes in roots and/or leaves were discussed.

#### 3.2.1. Genes Involved in Cell Wall Hydrolysis and Repair

The cell wall prevents excess Cd from entering the cytoplasm of plants [30]. Under Cd stress, enhanced cell wall biosynthesis has been found to improve Cd tolerance and accumulation in *A. thaliana* [31], rice [32], and high-Cd-accumulating pak choi genotype [33]. Notably, most genes encoding key enzymes of lignin biosynthesis, including phenylalanine amino lyase (*PAL*), cinnamate 4-hydroxylase (*C4H*), cinnamoyl CoA reductase (*CCR*), cinnamyl alcohol dehydrogenase (*CAD*), laccase (*LAC*) and *POD* were induced by Cd treatment in roots and leaves, indicating that lignin is induced to bind more Cd in the cell walls of *C. gigantea*.

Xyloglucan endoglycosidases/hydrolases (XTHs) was a key enzyme to affect cell wall expansion [34]. XTH encoding genes have been found to be related to Cd accumulation in plants. For example, in *A. thaliana*, *XTH33* is a necessary gene for Cd accumulation in roots [35]. In this study, a large number of *XTH* genes were induced by Cd treatment in leaves, indicating that these genes may regulate cell wall structure to affect Cd accumulation in leaves of *C. gigantea*.

#### 3.2.2. Potential Metal or Metal Complex Transporter Encoding Genes in Different Tissues

Cd is a nonessential metal for plants, and no specialized ion channels and transporters have been discovered for Cd. Cd mainly enters plants from roots through the transport carriers of other divalent cations such as manganese (Mn^2+^), iron (Fe^2+^), calcium (Ca^2+^), and zinc (Zn^2+^) [36]. Different transporter members, such as ATP-binding cassette transporters (ABC transporter) [37], cation diffusion facilitators (CDF) [38], and cation exchangers (CAX), can help plants resist Cd stress. In this study, several genes related to metal ion transporters, such as *IRT1* (Asia05G001745), *COPT1* (Asia07G000273), and *ZIP5* (Asia03G002673) were upregulated in leaves, indicating that these genes are likely involved in Cd transport. Connolly et al. (2002) found that Arabidopsis is more sensitive to Cd after overexpression of *AtIRT1*, indicating that *AtIRT1* can not only maintain the metabolic balance of iron in Arabidopsis, but also participate in the transport of Cd [39]. Gao et al. (2014) analyzed the transcriptome of *Sedum alfredii* and found that Cd stress greatly induced the expression of *SaZIP*1 in roots and leaves, and the expression level of *SaZIP*1 in the Cd hyperaccumulation ecotype *S**. alfredii* was nearly 100 times higher than that in non-Cd hyperaccumulation ecotype plants [40]. Chou et al., (2011) found that the expression level of *OsZIP3* in rice roots under Cd treatment significantly increased. They believe that *OsZIP3* is involved in the absorption of Cd in rice roots and the transportation of Cd from roots to shoots [41]. Therefore, we infer that *ZIP5* (Asia03G002673) is involved in Cd accumulation in *C. gigantea*.

Meanwhile, we found that three *DTX* genes were differentially expressed in leaves after Cd treatment. *DTX* family is a multifunctional gene family that participate in many processes in plants [42]. Overexpression of cotton *DTX* gene in Arabidopsis can enhance the drought, salt and cold resistance of transgenic Arabidopsis [43]. *DTX* transporters are also involved in the transport of secondary metabolites, such as alkaloids, flavonoids and some hormones. In addition, DTX is also involved in the elongation of hypocotyl cells. It can be considered that the *DTX* gene has diverse functions in *C. gigantea*.

In summary, we consider that these transporters may play a key role in Cd absorption or compartmentalization in *C. gigantea*. Further study of their precise molecular functions may be helpful to reveal different Cd response mechanisms of roots and leaves.

#### 3.2.3. Genes Involved in Cd Chelation Protein Encoding

When Cd enters cells, small molecular substances, including organic acids, amino acids, metallothionein (MT), and plant-chelating peptides, in the cytoplasm can binding Cd to form complexes to reduce Cd toxicity. MTs are ubiquitous metal-binding proteins with low molecular weights that are rich in cysteine and can be induced by metals [44]. MTs are not only involved in metal balance and detoxification in organisms, but also have a stronger ability to prevent oxidative stress [45]. In this study, three MT encoding genes, *MTB2* (Asia02G001361), *MTB3* (MSTRG.1816), and *MTB15* (Asia07G002256) (Appendix A) were upregulated in the leaves of *C. gigantea* under Cd treatment, which were consistent with many previous studies [46,47,48,49,50]. These results suggest that MTs contribute to Cd detoxification in *C. gigantea*.

Defensin is a cationic short peptide that widely exists in animals and plants and has a variety of defense functions. The defensin genes *AtPDF2.5* and *AtPDF2.6* in Arabidopsis were found to play an important role in the Cd response [51]. In vitro yeast experiments also showed that this gene can significantly enhance yeast resistance to Cd and revealed that *AtPDF2.5* per molecule can bind two Cd ions [52]. In rice, defensin unloaded Cd from the cytoplasm and entered the xylem to participate in long-distance transport by chelating Cd and secreting it outside the cell membrane [53]. In this study, an upregulated defensin encoding gene (MSTRG.14905) in the leaves of *C. gigantea* under Cd treatment (Appendix A) may be involved in Cd chelation to affect Cd transport and detoxification, but further experiments are needed to determine the role of this gene.

#### 3.2.4. Antioxidant Enzymes or Proteins Encoding Genes in Different Tissues

Heavy metal stress can lead to abundant production of reactive oxygen free radicals, which can damage the plant cell membrane. Plants can produce a protective system to eliminate the generated free radicals to reduce stress. This protective system consists of nonenzymatic antioxidants, such as TRXs, glutathione (GSH), ascorbic acid (ASA) and antioxidant enzymes include superoxide dismutase (SOD), POD, and catalase (CAT) [54]. In this study, several POD encoding genes were downregulated in the roots of *C. gigantea*, whereas many genes related to antioxidant enzymes, including SOD, POD, CAT and ascorbate peroxidase (APX) were upregulated in leaves (Appendix A). The results were consistent with the change in CAT activity in roots and leaves (Figure 2D). The induced antioxidant enzymes encoding genes suggest that they play important roles in regulating ROS accumulation under Cd stress.

In addition to antioxidant enzymes, TRXs and glutaredoxins (GRXs) are important antioxidant proteins in cells [55]. TRXs, which play a role in cellular redox systems by facilitating the reduction of other proteins via their dithiol-disulfide active site, reduce oxidative stress and other environmental stresses, protect proteins from oxidative aggregation and inactivation to promote protein folding, regulate apoptosis via denitrosylation, and modulate inflammation [47,56]. Like TRXs, GRXs are a kind of thioldisulfide oxidoreductases involved in the protection and repair of protein and nonprotein thiols under oxidative stress [57,58]. In the present study, one TRXs encoding genes (Asia04G000080), and two GRX encoding genes (Asia10G001567 and Asia07G000485) were upregulated in leaves (Appendix A), suggesting that they contribute to Cd tolerance in *C. gigantea* plants.

## 4. Materials and Methods

### 4.1. Seed Germination Experiment under Cd Stress

Seeds of *C. gigantea*, which were collected from Honghe, Yunnan Province, China, were sterilized using a 1% sodium hypochlorite solution for 5 minutes and washed with deionized water 3 times. The sterilized seeds were wrapped with gauze and soaked in sterilized water for 24 hours (h) to break dormancy. For the seed germination test, a gradient of Cd concentrations (i.e., 5, 10, 15, 20 and 30 mg L^−1^ Cd^2+^ supplied by CdCl_2_•2.5H_2_O) was set, 5 mL of which was added to the petri dishes covered with double-layered filter papers. Ten seeds of the same size were germinated in each petri dish and three repetitions were set for each Cd concentrations. Then, the dishes were placed in a lightproof box and cultured at constant temperature (28 °C). Germination was observed and recorded at 9:00 every morning. The germinated seeds were defined as seeds with at least 1-mm hypocotyls and radicles. The experiment ended when no new seeds germinated. The root lengths of ten germinated seeds in a dish were randomly measured using a vernier caliper at the end of the experiment. The germination rate and tolerance index (TI) were calculated to assess the Cd tolerance of seed germination of *C. gigantea* in accordance with the following formula [59]:

Germination rate = number of germinated seeds/total number of seeds × 100% 

Tolerance index = average length of roots under Cd treatment/average length of roots under control condition × 100%

### 4.2. Plant Materials and Cd Treatment

Sterilized seeds were planted in flowerpots (three seeds per pot) and placed in an artificial greenhouse at a temperature of 28 ± 3 °C. The pots were regularly watered to keep the soil moist. When the seedlings grew to approximately 20 cm height, the seedlings were divided into two groups. In the seed germination experiment, it was found that the concentration of 30 mg L^−1^ led to seeds suffering moderate to severe stress, while still being able to germinate, so concentration was set as 30 mg L^−1^. The treatment group was irrigated with 100 ml of 30 mg L^−1^ Cd^2+^ solution, and the control group was irrigated with 100 ml of sterile water. After Cd treatment for 24 h, the roots and leaves were separately harvested for physiological and transcriptomic analysis. Roots were first soaked with Na_2_EDTA (15 mM) for 15 min and then washed with deionized water three times [60]. Samples were frozen in liquid nitrogen and immediately stored at −80 °C for subsequent measurements. Nine samples from three pots were collected as three biological replicas.

### 4.3. Detection of Cd Accumulation, H_2_O_2_ Concentrations, and Catalase (CAT) Activity

The roots and leaves of each treatment were separated and dried in a 55°C oven to constant weight (at least 48 h). The dried root and leaf samples were microwave-digested with HNO_3_, and Cd concentrations were measured by ICP–MS [61]. H_2_O_2_ concentrations and CAT activity were detected using Solarbio kits according to the instructions (Beijing Solarbio Technology Co., Ltd., Beijing, China). The CAT enzyme activity was measured at 240 nm at the beginning of the reaction and 1 min later (Infinite M200PRO). The H_2_O_2_ content was measured at 415 nm, and 1 μmol H_2_O_2_ degradation per gram of tissue per minute was defined as a CAT activity unit. The significance of the differences between different treatments and different tissues were analyzed using GraphPad 8.0.

### 4.4. RNA Extraction and RNA Library Construction

RNA was extracted from roots (CR) and leaves (CL) of the control and roots (TR) and leaves (TL) treated with Cd by the TRIzol method (Invitrogen, CA, USA). First, RNA concentrations and integrity were detected with an Agilent 2100 (Agilent Technologies, Palo Alto, CA, USA), and RNA degradation and contamination were detected by 1% agarose gels without nuclease. Qualified high-quality RNA was used for the construction of the RNA-Seq library (refer to the library construction instructions ofGeneDenovo Biotechnology Company, Guangzhou, China.). Finally, the qualified RNA-Seq library was sequenced on Illumina NovaSeq6000. All of the raw RNA-Seq data have been deposited in the SRA database under NCBI (Accession No. PRJNA810329).

### 4.5. Data Filtering and Mapping 

Raw data obtained from Illumina underwent a series of operations to obtain high-quality clean data. First, the adapters connected during sequencing were removed. Next, reads with N content exceeding 10% were removed, and low-quality reads with Q-value ≤ 20 were deleted. The obtained high-quality clean data were mapped to the reference genome of *C. gigantea* measured by our team (not yet published) using HISAT 2.2.4.

### 4.6. Identification of Differentially Expressed Genes (DEGs)

In this study, gene expression levels were estimated using String Tie v1.3.1 software [62]. The expression level of the transcript was measured using the FPKM value. The DEGs were defined as those with a fold change ≥ 2 and false discovery rate (FDR) < 0.05.

### 4.7. Go and KEGG Enrichment Analysis

All DEGs were compared to the Gene Ontology database and relevant pathway database, the gene number of each term and pathway was calculated, and then a hypergeometric test was used to determine the GO terms and pathways significantly enriched in DEGs compared with the whole genome background. FDR < 0.05 was defined as the threshold for significant enrichment.

### 4.8. Gene Expression Validation

Twelve DEGs identified in both roots and leaves were randomly selected, and the RNA used for transcriptome sequencing was reversed transcribed to cDNA according to the protocol of the TransScript All-in-First-Strand cDNA Synthesis SuperMix for qRT-PCR Kit. The synthesized cDNA was used as a template for qRT-PCR. Specific primers were designed on the Integrated DNA Technologies website (https://sg.idtdna.com/, 30 August 2021), and specific primer information used in this study is listed in Appendix A. According to previous research, *CYP23* (Asia05T000290) was used as the internal reference gene [63]. qRT-PCR was carried out using the Bio–Rad CFX96 fluorescence quantitative PCR System. The parameters of the reaction system and time designed by previous studies, the dissolution curve and the relative expression of each gene were all calculated [64].

### 4.9. Statistical Analysis

Statistical analyses were performed using the SPSS version 18.0. An independent-sample *t*-test was used to analyze significant differences between two pairs of samples. Bar charts, linear regression analysis, Venn diagrams, and GO and KEGG enrichment analysis were performed using the online platform Omicshare (https://www.omicsmart.com, 15 September 2021).

## 5. Conclusions

In the present study, Cd tolerance characteristics and the molecular mechanism of *C. gigantea* were explored to evaluate the phytoremediation potential of this species in Cd-contaminated soils. Seed germination results indicated that *C. gigantea* is able to grow with certain Cd concentrations. Biochemical and transcriptomic analysis showed differential Cd response characteristics between roots were and leaves of *C. gigantea*. It was specifically reflected that the roots of *C. gigantea* were subjected to oxidative stress, whereas the leaves of *C. gigantea* activated several Cd detoxification processes to cope with Cd, including (1) upregulation of transporter genes involved in Cd absorption, efflux, or compartmentalization, (2) induction of gene expression involved in cell wall remodeling, (3) enhancement of the antioxidant system to control ROS accumulation, and (4) induction of chelation encoding genes. Nevertheless, with the increase of Cd treatment time, the growth of *C. gigantea* was not significantly inhibited, which may be attributed to the transfer of Cd from roots to aboveground parts. This study provides important information about *C. gigantea* response to Cd, which is useful for evaluating the potential of *C. gigantea* in the remediation of Cd-contaminated soils. However, Cd accumulation and transport characteristics in *C. gigantea* need to be further understood. Moreover, more comprehensive and clear Cd response mechanisms in *C. gigantea* needs to be further studied.

## Figures and Tables

**Figure 1 ijms-23-03329-f001:**
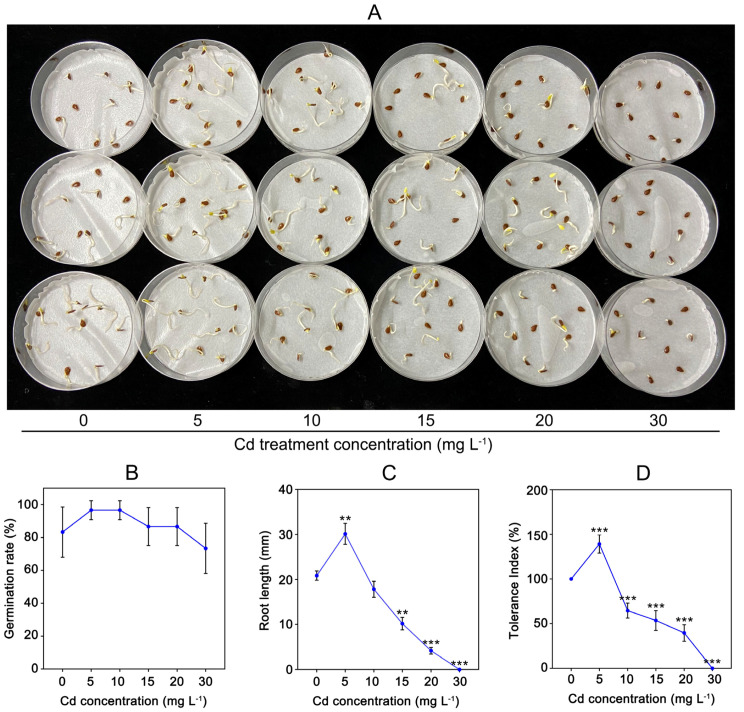
The effect of Cd concentrations on seed germination of *C. gigantea*. (**A**) The germination of *C. gigantea* seeds under different Cd concentrations treatment. (**B**) Seed germination rate under different Cd concentrations. (**C**) The root length under different Cd treatments. (**D**) Tolerance index under different Cd concentrations. Data represent the means ± standard deviations (n = 3); independent sample t test is performed between Cd treatment and control samples; ** 0.001 < *p* < 0.01, *** *p* < 0.001 (**B**–**D**).

**Figure 2 ijms-23-03329-f002:**
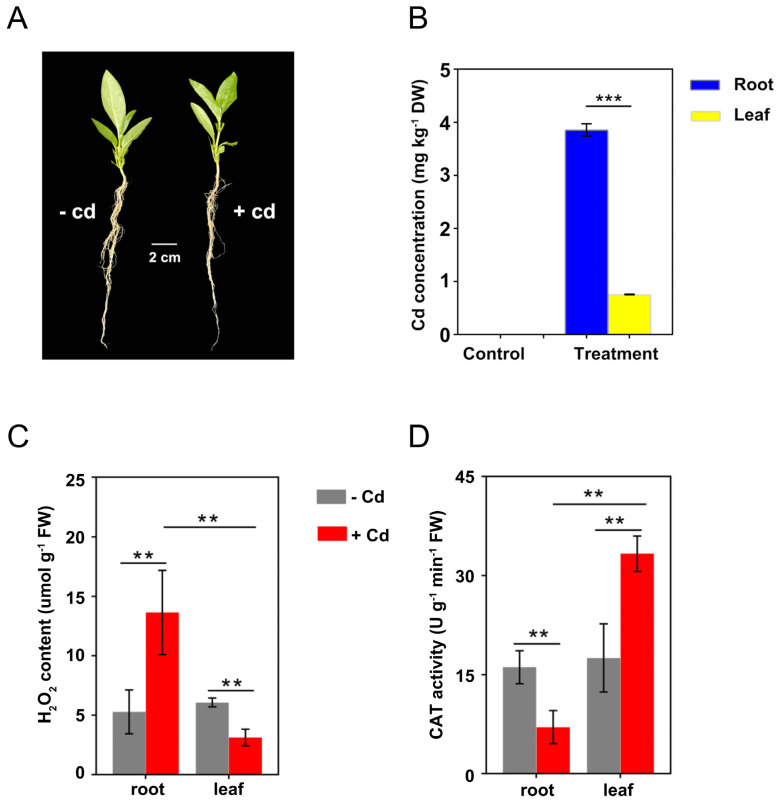
Cd accumulation and physiological responses of *C. gigantea* under Cd treatment. (**A**) Phenotype of *C. gigantea* seedlings treated with and without 30 mg L^–1^ Cd (100 mL) for 24 h. (**B**) Cd concentrations in roots and leaves of *C. gigantea*. (**C**) H_2_O_2_ concentrations in roots and leaves of *C. gigantea*. (**D**) CAT activities in roots and leaves of *C. gigantea*. One unit of CAT activity was defined as the quantity of enzyme degrading 1 μmol H_2_O_2_ per minute at 25 °C per gram of tissue. Data represent the means ± standard deviations (n = 3); independent sample t test is performed between two pairs of samples; ** 0.001 < *p* < 0.01, *** *p* < 0.001 (**B**–**D**). FW: fresh weight.

**Figure 3 ijms-23-03329-f003:**
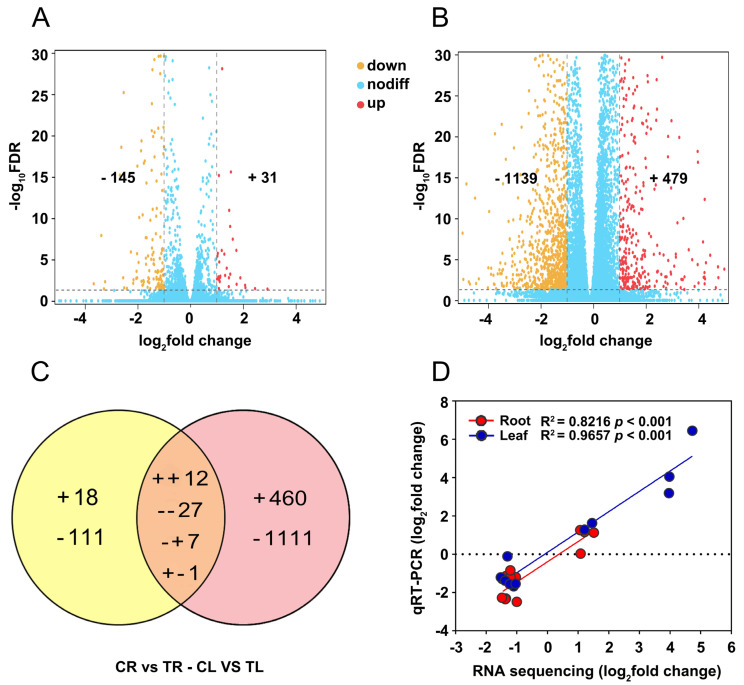
Differentially expressed genes (DEGs) in roots and leaves of *C. gigantea* and qRT-PCR validation. (**A**) Volcano plot of DEGs in in roots. (**B**) Volcano plot of DEGs in leaves. (**C**) Venn diagrams showing the unique and shared regulated genes in roots and leaves under Cd stress. (+, upregulated; -, downregulated). (**D**) Linear regression analysis of change folds for the 12 selected genes between RNA sequencing and qRT-PCR results in roots and leaves.

**Figure 4 ijms-23-03329-f004:**
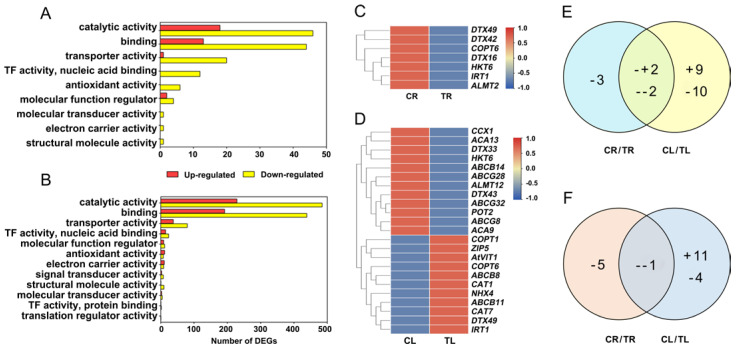
Enriched GO terms of differentially expressed genes (DEGs) and expression characteristics of DEGs related to Cd transport in roots and leaves of *C. gigantea*. (**A**) Number of DEGs of enriched GO terms based on molecular function in roots. (**B**) Number of DEGs of enriched GO terms based on molecular function in leaves. (**C**) Heat map showing expression changes of the DEGs related to Cd transport in roots. (**D**) Heat map showing expression changes of the DEGs related to Cd transport in leaves. (**E**) Common and specific DEGs related to Cd transport between roots and leaves. (**F**) Common and specific DEGs related to antioxidant process between roots and leaves. (+, up-regulated; -, down-regulated).

**Figure 5 ijms-23-03329-f005:**
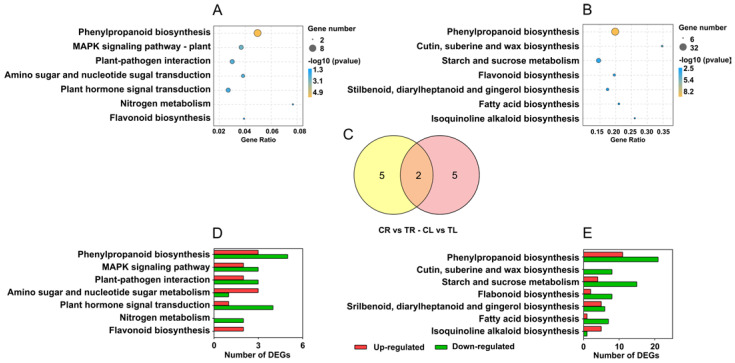
Enriched KEGG pathways of differentially expressed genes (DEGs) in roots and leaves of *C. gigantea*. (**A**) Enriched KEGG pathways of DEGs in roots. (**B**) Enriched KEGG pathways of DEGs in leaves. (**C**) The number of common and specific enriched KEGG pathways of DEGs in roots and leaves. (**D**) Number of up- and down-regulated genes of the enriched KEGG pathways in roots. (**E**) Number of up- and down-regulated genes of the enriched KEGG pathways in leaves.

## Data Availability

All data were included in the manuscript.

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
