# Peer review of "Comparative Transcriptomics Analysis of Roots and Leaves under Cd Stress in Calotropis gigantea L."

_ijms, 2022, doi:10.3390/ijms23063329_

Round 1
Reviewer 1 Report
The discussed manuscript "Comparative transcriptomics analysis of roots and leaves under Cd stress in Calotropis gigantea L." it is a very interesting work that touches upon important topics. The manuscript is written in a logical way, contains all the essential elements and should be presented to a wider group of scientists.
Due to the protection of the environment, which is already polluted to a large extent, it is very important to research the aim of which is to understand the detailed mechanisms of plant reactions to individual pollutants. The submitted manuscript addresses issues related to the influence of cadmium on C. gigantea. This is very important from the scientific point of view as it brings us closer to determining the exact mechanism of the influence of heavy metals on plants. Understanding the exact mechanism will allow us, on the one hand, to protect these plants, and on the other hand, it is possible to select such plant species so that they can be used for phytoremediation of contaminated areas, so that they can, for example, be reused for agriculture. In my opinion, the manuscript is written honestly. The results are clearly and properly presented and presented together with their statistical analysis.Author Response
Dear Reviewer:
Here is my reply to your comment :
Reviewer 1
1) The discussed manuscript "Comparative transcriptomics analysis of roots and leaves under Cd stress in Calotropis gigantea L." it is a very interesting work that touches upon important topics. The manuscript is written in a logical way, contains all the essential elements and should be presented to a wider group of scientists.
Due to the protection of the environment, which is already polluted to a large extent, it is very important to research the aim of which is to understand the detailed mechanisms of plant reactions to individual pollutants. The submitted manuscript addresses issues related to the influence of cadmium on C. gigantea. This is very important from the scientific point of view as it brings us closer to determining the exact mechanism of the influence of heavy metals on plants. Understanding the exact mechanism will allow us, on the one hand, to protect these plants, and on the other hand, it is possible to select such plant species so that they can be used for phytoremediation of contaminated areas, so that they can, for example, be reused for agriculture. In my opinion, the manuscript is written honestly. The results are clearly and properly presented and presented together with their statistical analysis.
Res: Thank you for reviewing our manuscript efficiently. It’s my honor to have your favorable comments. We have further improved the manuscript. We hope the revised manuscript is now acceptable for publication in International Journal of Molecular Sciences and look forward to hearing from you at your earliest convenience.
Reviewer 2 Report
In the manuscript at hand Jingya Yang and colleagues describe differential gene expression analyses the conducted on Calotropis gigantea - a fiber and medical plant also used as roadside trees - under Cadmium exposure. Aim of this study was to characterize molecular phenomena that occur as part of heavy metal tolerance and accumulation.
The study seems well conducted and the manuscript well written I would support its publication if some key questions/issues are addressed by the authors.
Major issues:
1) As the authors state in the manuscript that they used a custom genome assembly and genome annotation for their analysis. I have seen that there is already a genome published https://www.ncbi.nlm.nih.gov/labs/pmc/articles/PMC5919723/. If the authors use their own unpublished genome, a parallel analysis of the published genome must be performed to allow a fair assessment of the quality of the unpublished genome.Â
2) The description of the method is inadequate. Both software and library version numbers and full parameter disclosure need to be provided. In addition, RSEM gives TPM and not FPKM values. Which model was used for differential expression analysis?
3) Raw RNA-Seq data should be deposited at GEO/SRA.
Minor issues:
4) All figure legends: What statistical tests were performed (plus parameters)?, what groups were compared? what is N?
5) Fig. 3a/b axis labels missing.
6) Gene expression validation by qPCR on the same sample material is meaningless, can be omitted.
Â
Author Response
Dear Reviewer,
Thank you for reviewing our manuscripts efficiently. You gave us several insightful comments to greatly improve the quality of our manuscript.Â
The manuscript has been carefully revised according to your feedback and further revisions. We hope the revised manuscript is now acceptable for publication in International Journal of Molecular Sciences and look forward to hearing from you at your earliest convenience.
Yours sincerely
Dr. Jingya Yang

Round 2
Reviewer 2 Report
The authors have answered my objections to my satisfaction.
